# EvIL: Evolution Strategies for Generalisable Imitation Learning

## Abstract

We present Evolutionary Imitation Learning (EvIL), a general approach to imitation learning (IL) able to predict agent behaviour across changing environment dynamics. In EvIL, we use Evolution Strategies to jointly meta-optimise the parameters (e.g. reward functions and dynamics) fed to an inner loop reinforcement learning procedure. In effect, this allows us to inherit some of the benefits of the inverse reinforcement learning approach to imitation learning while being significantly more flexible. Specifically, our algorithm can be applied with any policy optimisation method, without requiring the reward or training procedure to be differentiable. Our method succeeds at recovering a reward that induces expert-like behaviour across a variety of environments, even when the environment dynamics are not fully known. We test our method's effectiveness and generalisation capabilities in several tabular environments and continuous control settings and find that it outperforms both offline approaches, like behavioural cloning, and traditional inverse reinforcement learning techniques.

## 1 Introduction

Imitation Learning - the problem of learning to imitate an agent's behaviour - has been successfully applied to many real world settings, including autonomous driving (Codevilla et al., 2018) and robotics (Yu et al., 2018). In this framework, the agent's goal is to reproduce the behaviour of an expert agent, as shown in a set of given demonstrations.

Broadly speaking, we can divide imitation learning into three families: *offline* methods like behavioral cloning (BC) (Pomerleau, 1988), *online* methods like inverse reinforcement learning (IRL, (Ziebart et al., 2008a)), and *interactive* approaches like DAgger (Ross et al., 2011). BC is one of the simplest IL methods, where supervised learning is used to predict expert actions from expert observations.

Although more sophisticated forms of BC exist, few focus on the ability to predict expert behaviour across *changing environment dynamics*. As a policy is trained, it becomes inextricably tied to the underlying environment, and therefore fails to accurately predict the expert's behaviour in a new or changing environment. This makes the BC framework fundamentally limited. We are interested in the ability of predicting an expert's behaviour, in an environment different from the one where the trajectory data was collected.

In this setting, Inverse Reinforcement Learning (IRL) methods are often the best approach. IRL aims to recover a reward function under which a given agent is optimal. While environment dynamics frequently change, agents' intentions remain more constant, making reward functions the most succinct and transferable description of agent behaviour (Ng et al., 2000). IRL has been successfully used to predict agent behaviour across a variety of different settings, such as trying to predict customer response to changes in the economic climate (Rust, 1994), pedestrian navigation (Kitani et al., 2012) or taxi-cab driving (Ziebart et al., 2008a).

IRL presents an alternative approach towards IL: rather than using demonstrations to recover environment *dependent* behaviours directly, the goal is to recover the *invariant* reward functions. These can in turn be used to produce *co-variant* behaviours under novel environment conditions, e.g. by using reinforcement learning or other optimisation tools. This recipe has repeatedly produced practical successes (e.g. in robotics (Silver et al., 2010; Ratliff et al., 2009; Kolter et al., 2008; Ng et al.,

2006; Zucker et al., 2011), computer vision (Kitani et al., 2012), and human-computer interaction (Ziebart et al., 2008b; 2012)).

While successful, traditional IRL methods suffer from several key limitations: 1. Most widely used methods are fundamentally adversarial in nature, which means that careful tuning of learning rates or update frequencies can be required to elicit strong policy performance (Barde et al., 2020). 2. Access to the underlying dynamics or a good simulator is generally assumed, therefore application remains challenging in settings where interaction is potentially unsafe, expensive, or good models do not exist (e.g. a robot manipulating a deformable object like a tomato) and 3. The reward model loss function has to be differentiable, limiting the ability to optimise complex objectives (e.g. minimising the number of interactions required to match expert behaviour).

To address these shortcomings, we propose *Evolutionary Imitation Learning* (EvIL), a *general* IL framework able to optimise any non-differentiable objective function, or combination of, while recovering both reward and (optionally) environment dynamics. As a model of environment dynamics is trained directly to induce a behaviour similar to the experts', we circumvent the objective mismatch issue (Farahmand et al., 2017) common to other offline, Model-Based RL approaches.

As a result, EvIL can easily be applied to settings where the original expert training environment is either not provided or underspecified. This is an extremely common setting, as datasets of real-world applications of IL are often openly available while the data collection environment is not (e.g. robotics, autonomous driving). In settings where we have some previous knowledge of either the transition function (e.g. through a simulation) or the reward dynamics. EvIL offers a flexible framework able to incorporate this known information into the optimisation process.

EvIL uses a bi-level optimisation process: the **outer loop** generates training parameters passed to the inner loop. In our implementation the **inner loop** uses RL to train a policy with the parameters provided and the outer loop uses a cross-entropy loss to measure the quality of the fit between the resulting RL policies and the demonstration data. In the outer loop, we use Evolution Strategies (Salimans et al., 2017), a sample based gradient estimation approach, to estimate the gradient of the loss *through the policy optimisation procedure* and update the outer loop parameters accordingly. This methods effectively generates a policy that imitates the expert, while recovering information about both the original agents' intents and the underlying environment dynamics. Crucially, this information is sufficient for computing adapted policies under novel scenarios.

A naive implementation of this method, however, either fails to imitate expert behaviour during meta training or results in reward functions that don't generalise to novel settings. The first problem is that estimating the ES gradient through many episodes of RL training leads to *vanishing gradients* and therefore a loss landscape which is hard to optimise. We address this by gradually increasing the number of RL inner loop steps, which not only makes the optimisation more robust, but also reduces computational cost. To address the issue of reward functions not generalising, we introduce a regularisation regime which biases the reward function to be as *invariant* as possible to the input.

As jointly optimising the two objectives (regularisation and BC-loss) is difficult for ES, we introduce a two stage optimisation procedure. We first update the parameters according to the ES gradient, that optimises for a BC minimum norm solution close to the ES solution. This is effectively equivalent to a distillation step. In our ablation experiments we verify that both technical contributions are required to achieve good performance.

We test our method in a Gridworld environment as well as classic control tasks like Reacher (Lenton et al., 2021) and Pointmass. In all test environments, our recovered rewards successfully generalise and generate agents better able to imitate the expert than previous methods. Additionally, when provided with an *underspecified* environment or no environment at all, our model is able to correctly recover the missing environment parameters (or a full transition function) while also recovering the reward function. In the case of underspecified dynamics, the reward is also generalisable to novel environment dynamics. We will make all the code used in this paper open source upon acceptance.

## 2 BACKGROUND AND PROBLEM SETTING

We assume a Markov Decision Process (MDP) (Puterman, 1994) parameterised by $\langle \mathcal{S}, \mathcal{A}, T, T_0, R, \lambda, H \rangle$ where $\mathcal{S}, \mathcal{A}$ are the state and action spaces, $T(s_{t+1}|s_t, a_t)$ is the tran-

sition function, $T(s_0)$ is the initial state distribution, $R(r_{t+1}|s_t, a_t, s_{t+1})$ is the reward function (where $r_{t+1} \in [-1, 1]$), $\lambda$ is the discount factor and $H$ the time horizon.

In our setting, we assume that across tasks, the transition function $T$ might change but everything else, including $R$, stays the same. In the standard RL setting we want to maximise

$$\mathcal{J}(\pi) = \mathbb{E}_{T_0, T, \pi}[\sum_{t=0}^{H-1} \lambda^t R(r_{t+1}|s_t, a_t, s_{t+1})]$$

In the IL setup we have access to agent trajectories $\mathcal{D}_E \sim \{\tau_0, ..., \tau_N\}$ where $\tau_E = (s_0, a_0, ..., s_H, a_H)$ and we want to recover the policy that generated those trajectories $R$. BC commonly uses a cross entropy loss to optimise the learner policy $\pi_L$, i.e.:

$$\arg \min_\pi \mathcal{L}(\pi) = -\frac{1}{N} \sum_{t=1}^{H} \log \pi_L(a_t|s_t)$$

### 2.1 EVOLUTION STRATEGIES

Evolution Strategies are population-based stochastic optimization algorithms that use random noise to generate a population of candidate solutions. These solutions are then evaluated using a fitness function and the population is iteratively improved over time by assigning higher weight to better-performing population members. This causes the population to move closer and closer to the optimal solution, and the process is repeated until a satisfactory solution is found. Recently, ES has been successfully applied to a variety of tasks (Real et al., 2019; Salimans et al., 2017; Such et al., 2018). ES algorithms are gradient-free and well-suited for (meta-) optimisation problems where the objective function is noisy or non-differentiable and the search space is large or complex (Beyer, 2000; Lange, 2023; Lu et al., 2023; 2022; Houthooft et al., 2018). This includes reward function shaping (Niekum et al., 2010) and RL hyperparameter search (Elfwing et al., 2018). There are several types of ES algorithms, one of the most well known is the covariance matrix adaptation evolution strategy (CMA-ES) (Hansen & Ostermeier, 2001), which represents the population by a full-covariance multivariate Gaussian. Although CMA-ES can be applied to our problem, it has only proven successful in low to medium dimension optimisation spaces. Another widely applied ES algorithm is OpenAI-ES (Salimans et al., 2017) which estimates the gradient through the following function:

$$\nabla_\theta \mathbb{E}_{\epsilon \sim N(0,1)} F(\theta + \sigma\epsilon) = \frac{1}{\sigma} \mathbb{E}_{\epsilon \sim N(0,1)} \{F(\theta + \sigma\epsilon)\epsilon\}$$

This is an unbiased estimate and, in contrast to meta-gradient approaches, ES avoids the need to backpropagate the gradient through the whole training procedure, which often results in biased gradients due to truncation (Werbos, 1990; Metz et al., 2022; Liu et al., 2022).

## 3 RELATED WORK

### 3.1 IMITATION LEARNING

Our approach straddles the gap between offline and online methods: we do not assume access to the environment (as offline methods do), but search over reward functions rather than $Q$-functions (as online methods do). If no access to the environment is provided at train time, our method cannot guarantee robustness to compounding errors (Swamy et al., 2021), but still inherits some of the benefits of reward-matching methods like IRL.

### 3.2 INVERSE REINFORCEMENT LEARNING

IRL is commonly framed as a two-player zero-sum game between a policy player and a reward function player (Swamy et al., 2021). Intuitively, the reward function player tries to pick out differences between the current learner policy and the expert demonstration, while the policy player attempts to maximise this reward function to move closer to expert behaviour. As pointed out by Finn et al. (2016), this setup is effectively a GAN (Goodfellow et al., 2014) in the trajectory space. On tabular problems, one can solve this game by having both players follow a no-regret strategy like multiplicative weights (Syed & Schapire, 2007) or by having a no-regret vs. a best-response dynamic

(Ziebart et al., 2008a). Our approach fits into this latter family as we compute a best response by optimizing the current reward function via reinforcement learning. Of course, once we move out of the tabular regime, we need to use function approximators like deep networks to represent both our reward function and policy, which is common in the prior work (Ho & Ermon, 2016; Fu et al., 2018; Wulfmeier et al., 2016).

The key difference between our work and the prior work is that we can pick reward function discriminators based on non-differentiable objectives. In traditional IRL, we usually use a "performance difference" objective (i.e. $\ell(r) = J(\pi_E, r) - J(\pi, r)$) that is linear in the reward function and therefore differentiable (Ziebart et al., 2008a; Swamy et al., 2021). However, there are a variety of objectives we could care about that we can't cleanly write down as differentiable function of the reward. For example, in response to the well-established computational inefficiency of IRL (Swamy et al., 2023), we might want to optimise for reward functions that, while differentiating between the learner and the expert, are shaped to ensure efficient policy optimisation. Our ES-based framework allows us to optimise these auxiliary objectives and therefore is significantly more flexible and general than the prior art.

### 3.3 Model-Based Reinforcement Learning

By learning a model from collected data and then planning in it, model-based reinforcement learning approaches can be far more sample-efficient than their model-free counterparts (Hafner et al., 2023; Schrittwieser et al., 2020). However, model-based RL approaches typically assume online access to the environment or access to a large offline dataset to fit a model that produces accurate simulated rollouts for the learner. As we operate in the offline setting and do not assume full coverage of the expert data, we cannot directly apply these approaches.

A key concern with any approach that fits a model is how the training error of the model translates to the quality of the policies learned by acting in it. Theory tells us that in the worst case, we need to be close in an $\ell_\infty$ sense to the ground truth dynamics to be able to accurately evaluate an arbitrary policy (Kearns & Singh, 2002). Unfortunately, there is no known way to guarantee this, so in practice we often resort to minimizing a simple loss function (e.g. MSE on the next-step prediction). However, this means that we can no longer guarantee that a policy that is optimal in our model will perform well at test time, an issue termed *objective mismatch* in the MBRL literature (Farahmand et al., 2017; Lambert et al., 2020). Theoretically-grounded approaches to fix this issue require online interaction or adversarial training (Vemula et al., 2023). In contrast, because our approach directly optimises a model that induces expert-like behaviour, we are able to circumvent the objective mismatch issue entirely. Our approach requires fewer assumptions than other IRL approaches that also optimise an environment model (Reddy et al., 2018; Herman et al., 2016).

## 4 Method

IRL is frequently conceptualised as having an *outer loop* (in which a reward function is chosen via minimising a classification loss) and an *inner loop* (in which the reward function is maximised over the horizon by a reinforcement learning algorithm). At a high level, our method replaces the outer-loop first-order supervised learning step with an zeroth-order evolutionary update. This allows us to a) optimise non-differentiable objectives and b) optimise more than just the reward function. More explicitly, our bi-level optimization problem has the following form:

- The **outer loop**'s goal is to propose a set of parameters for the inner loop training. In traditional IRL, this is just a reward function $R$. In our framework, we can propose additional components, like a set of transition dynamics $T$ or hyperparameters for the inner loop RL algorithm. It does this by minimising some (not neccesarily differentiable) loss function $\mathcal{L}$.

- The **inner loop**'s goal is solving the RL problem, using the MDP and hyperparameters defined in the outer loop. This is identical to standard IRL.

---

**Algorithm 1** EvIL

---

**Input:** Trajectories $\tau_E$ from expert, learning rate $\alpha$, noise standard deviation $\sigma$
**Output:** Trained policy $\pi$, learned reward $R_\theta$, transition function $T_\phi$
Initialise policy $\pi$ and parameters $\theta, \phi$, population size $N$, $\ell_1$ coefficient $\beta$
**repeat** ▷ Outer-loop optimisation
    Generate Gaussian noise $\epsilon_1, ...\epsilon_N \sim \mathcal{N}(0, I)$ to generate $N$ members in the population
    **for** $i = 0, ..., N - 1$ **do**
        $(R_{\theta_i}, T_{\phi_i}) = (R_\theta, T_\phi) + \sigma\epsilon_i$
        $\pi_i \leftarrow$ policy optimisation for $R_{\theta_i}$ under $T_{\phi_i}(s_{t+1}|s_t, a_t)$      ▷ Inner-loop optimisation
        Calculate $\mathcal{L}_i = -\mathbb{E}_{(s_t,a_t)\sim\tau_E}[\log \pi_i(a_t|s_t)]$ for each policy $\pi_i$
    **end for**
    $(\theta, \phi)_{t+1} \leftarrow (\theta, \phi)_t - \alpha\frac{1}{N\sigma}\sum_{i=1}^{N}\mathcal{L}_i\epsilon_i$      ▷ Estimate gradient and update meta parameters
    $\theta_0^D \leftarrow \theta_{t+1}$
    **for** $i = 0, ..., M - 1$ **do**      ▷ Distillation Loop
        $\theta_{i+1}^D \leftarrow \theta_i^D - \alpha\nabla_{\theta_i^D}\left(\mathbb{E}_{(s_t,a_t)\sim\tau_E}\left(R_{\theta_{t+1}}(s_t, a_t) - R_{\theta_i^D}(s_t, a_t)\right)^2 + \beta\ell_1(\theta_i^D)\right)$
    **end for**
    $\theta_{t+1} \leftarrow \theta_M^D$
**until** convergence

---

## 4.1 CHOICE OF FITNESS FUNCTION

We now detail two example *fitness functions* we can use to optimise the parameters for the inner loop which highlight the flexibility of our method over traditional IRL.

1. Assuming we don't have access to the environment the expert was acting, we would also need to estimate $T$. In essence, we want to search for an $(R, T)$ pair such that the induced optimal policy matches our demonstration data. We can jointly optimise over pairs by setting $\mathcal{L}(R, T) = -\mathbb{E}_{s\sim\tau_E}[\log \pi_{R,T}^*(a_t|s_t)]$ (i.e. the behavioural cloning loss), where $\pi_{R,T}^*$ is the optimal policy under $(R, T)$.

2. Even if we have access to the environment the expert was acting in, our framework enables optimisation for non-differentiable objectives like training time. For example, the moment-matching gradient with respect to *any* potential-based reward shaping term (Ng et al., 1999) is 0 (as it sums to 0 for any trajectory). This means that we could take the reward function returned by standard IRL and add to it a shaping term optimised by evolution to maximize the "area under the curve" of performance vs. environment interactions.

## 4.2 NETWORK DISTILLATION

Due to the resulting *sparsity*, minimum $\ell_1$ norm solutions are known to have better generalisation properties since they reduce dependency on potentially spurious features (Tibshirani, 1996). Accordingly, in traditional IRL, $\ell_1$ regularisation is commonly applied to the reward function to deal with finite-sample spuriosity, an idea with rigorous theoretical backing (Dudik et al., 2004). The naive option for including $\ell_1$ regularisation in EvIL is to simply add the regularisation term to our outer loop fitness function. However, this both requires ES to balance two different objectives (often unstable) and uses samples to estimate a gradient of a *differentiable objective*, which is inefficient.

To address this, we include the regularisation via a fully supervised *distillation* step: At each outer loop iteration $t$, we first apply the ES update from the *unregularised* fitness function to obtain a current reward function, $\mathcal{R}_{\theta_t}$. We next use a supervised learning step to fit this reward function on all expert trajectories with a different network, $\theta^D$ (initialised to $\theta_t$), applying $\ell_1$ regularisation to $\theta^D$. We finally use this updated $\theta^D$ as the new mean of the reward functions in the next outer loop.

This process uses ES to find *sufficiently complex* reward functions, while the distillation step ensures they are *as simple* (i.e. invariant) as possible. We find that this two-stage procedure is better at producing generalisable reward functions than directly including the $\ell_1$ loss as part of the fitness function. Please see Algorithm 1 for full details of our method, including the distillation step.

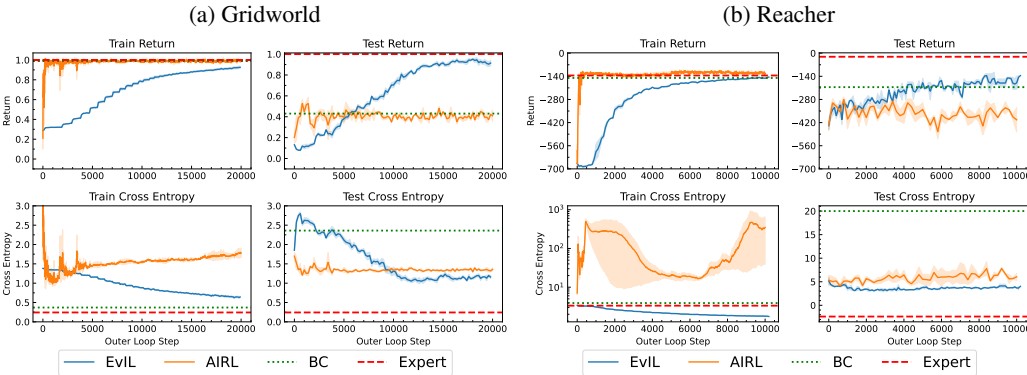

Figure 1: At the top, train and test returns. At the bottom, cross entropy loss. Reward on the x-axis is shown as fraction of expert reward, on the y-axis we have outer loop steps, or ES generations. Shading represents standard error.

### 4.3 POLICY RESETS AND INNER LOOP UPDATES

In the simplest setting, the agent is trained from scratch every time with a new variation of the reward function. In practice, this has two downsides: 1. it is **sample inefficient**, as the reward is only changing by a small amount, it is reasonable to continue training the previous policy. For long training regimes, this approach quickly becomes impractical. 2. Estimating the ES gradient through many episodes of RL training leads to **vanishing gradients** as shown in Figure 4a. Moreover, training agents through many episodes leads to higher noise in results, making the outer loop objective harder to optimise. The reward is not the only factor accounting for the final performance of an agent, as variables such as environment resets and action sampling also play a role. If we keep optimising the same inner loop policy $\pi$, we can minimise long data collection and interactions with the environment. We note that this "warm-starting" is standard in most practical implementations of IRL (Swamy et al., 2021; Ho & Ermon, 2016; Swamy et al., 2023).

### 4.4 RECOVERING ENVIRONMENT PARAMETERS

The EvIL framework can also be used in the offline setting, to recover information about the environment. EvIL is flexible: recovering a full transition function is possible, although it might not always be the best choice. Often, access to the dynamics might be partial, as if, for example, we had some robotic trajectories as well as access to a physics simulator. In this case, we prove we are able to successfully recover underlying information about the transition function not necessarily visible in the observation, such as the gravity variable in Cartpole (Kumar, 2020), or the position of an obstacle in Pointmass environment.

We represent the transition function $T(s_{t+1}|s_t, a_t)$ as a VAE (Kingma & Welling, 2022). To generate the starting state, we sample from the latent space $z \sim N(0, 1)$ and feed the vector through the VAE's decoder. The VAE's encoder takes in $p(z|s_t, a_t)$ and the decoder outputs $s_{t+1}$. If any prior information about the structure of the reward function is known, our optimisation process can co-learn thereward and transition function using the structural knowledge about either to optimise the other.

## 5 EXPERIMENTS AND RESULTS

In our experiments, we start by verifying that we can successfully recover reward functions in a fully online setting (i.e. with access to the true environment). We then train an agent with the recovered rewards on a test environment with different dynamics, and compare the performance to baselines. Afterwards, we analyse the framework's performance in a partially offline, and then fully offline setting (i.e. without environment access).

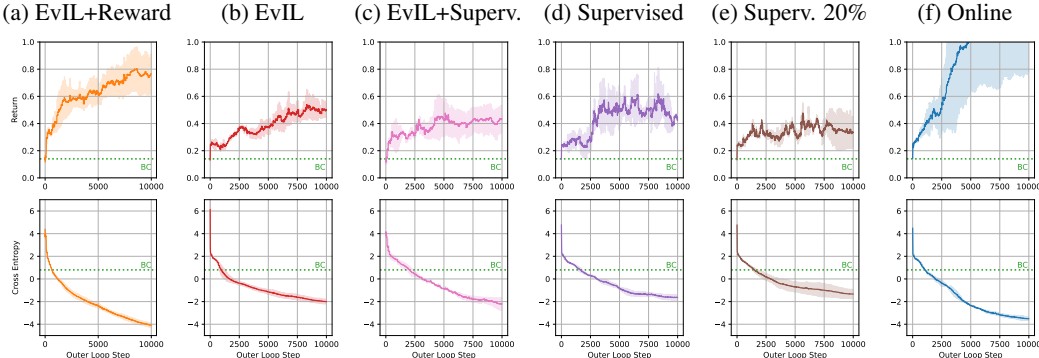

Figure 2: **Different strategies for optimising transition dynamics** All plots are recovering a reward function to match expert behaviour with EvIL. Each plot uses a different strategy for optimising the underlying transition function. The environment is Pointmass, and one trajectory is provided. The original agent obtains a return of 3. Shading represents standard error. **a.** Trains a transition function purely through optimisation of the BC loss with EvIL. The structure of the reward function is partially known. **b.** No knowledge of the reward function or its structure, the dynamics are optimised solely via EvIL. **c.** Dynamics are co-optimised through EvIL loss and MSE loss over provided demonstrations. **d.** EvIL Transition dynamics learned through supervised learning on the provided demonstrations. **e.** Only 20% of the data is used to train the transition function via supervised learning **f.** Online setting, environment is known

For all experiments, we report the Cross Entropy loss against the expert demonstration set as well as the average reward across many different initialisations of the environment. The latter helps to quantify how well the agent generalises to unseen train/test environment initialisations.

All our experiments are implemented in JAX (Bradbury et al., 2018) using the PureJaxRL Lu et al. (2022), Gymnax (Lange, 2022), and evosax Lange (2023) libraries to maximise parallelisation of training across ES population members.

## 5.1 REWARD RECOVERY - ONLINE SETTING

**Gridworld**: We first test our method in a 5x5 Gridworld environment with two different goals. The goals need to be found in the correct order to maximise return. The goal positions and agent's starting position are randomly chosen at the beginning of each episode. The environment observations are a one hot encoded representation of the Gridworld, where walls, different goals and the agent are encoded in different channels. We successfully recover the reward and expert's policy in the training environment without walls and use the reward to train a new agent in a new Gridworld environment with walls. We provide 100 expert demonstrations of length T=30. Figure 1a shows 5 seeds for the EvIL runs and 2 seeds for the Adversarial Inverse Reinforcement Learning (AIRL) baseline runs.

**Reacher**: We then test our method on the Reacher (Lenton et al., 2021) environment. We manage to correctly recover a generalisable reward that performs better than baselines on the test environment (where torque is increased by 10x). As in Gridworld, the goal and agent's starting position are randomly chosen at the beginning of each episode. We provide 50 expert demonstrations of length T=200. In Figure 1b we run 2 seeds for both the EvIL runs and the AIRL baseline.

For both environments, we match the AIRL baseline in terms of average reward in the train environment, but we outperform it according to all other metrics - return in test environment, as well as Cross Entropy in both train and test.

## 5.2 ENVIRONMENT AND REWARD RECOVERY

We analyse the performance of our method in the partially offline and fully offline setting in a simple Pointmass environment. The agent's goal is to move, in a continuous action space, towards a simple goal. When it gets close enough, the agent receives a reward and the agent position is reset to a random

position. The goal position is reset at the end of every episode (T=20). The observations include the agent and goal position at each time step.

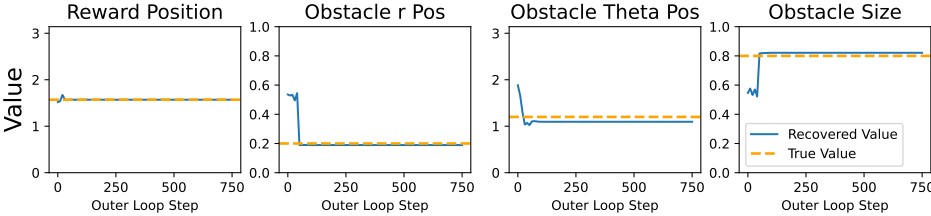

Figure 3: We successfully recover the goal position, obstacle position (in polar coordinates) and size

**Underspecified Environment** In this first setting, we have access to the real environment which has an obstacle of unknown size at an unknown position. In this experiment, we assume the goal position is static, and what the reward function needs to recover is the position of the goal. We first imitate expert trajectories from an environment where no obstacle is present. EvIL correctly chooses an obstacle with the minimum possible size value and on the border of the explorable space. If we instead try to imitate expert demonstrations where an obstacle was present, EvIL correctly recovers the position and size of the obstacle, as well as the correct reward position (Figure 3). We train on 20 trajectories using CMA-ES as the ES optimisation algorithm, as it converges faster in low-dimensional settings.

**Offline EvIL** Here, we assume no access or knowledge of the environment. All that is provided are the expert trajectories. We compare different approaches to generating our model, all shown in Figure 2 (2 seeds for each plot). All of them use EvIL to optimise a reward function. In this environment, we provide EvIL with 1 expert trajectory to imitate. We analyse the following settings and compare them to 2f (online):

**EvIL & some reward structure** 2a: The transition function parameters are randomly initialised, and then optimised directly by EvIL. However, we assume prior, partial knowledge of the reward function (i.e. we know it's a function of the distance between agent and goal).

**EvIL & no reward structure** 2b: As above, but no knowledge of the reward function is assumed.

**EvIL & Supervised Loss through ES** 2c: We don't assume any knowledge of the reward function, but we add an MSE loss for predictions over known expert trajectories. This is to try an encourage the model to predict realistic and interpretable transitions.

**EvIL on Supervised Model** 2d: The transition function is trained via MSE on the observed expert trajectories and used in the inner loop optimisation. It is not further optimised in the inner loop.

**EvIL on Partially Supervised Model** 2e: Same as above, but trained on a subset (20%) of the expert trajectory. Due to our very simple environment, even one trajectory is enough to train a good model, so we train this model on a subset of transitions to ensure we have a suboptimal model.

Overall, we observe that 1. All implementations vastly outperform the BC baseline 2. Knowledge of the reward function helps performance, indicating EvIL is using knowledge of the reward function shape to recover the transition function. 3. In 2c we notice, once again, that the ES procedure struggles to jointly optimise two different objectives (BC loss and MSE). This slows down convergence and hinders performance.

## 5.3 ABLATIONS

In our ablations we aim to answer the following questions: 1. What are the benefits of periodic distillation of the reward network with an analytical gradient vs adding a regularisation penalty to the ES objective? 2. What are the benefits of gradually increasing the number of inner loop steps vs always fully training the agent in the inner loop? 3. When "warm-starting" is applied, how does the number of epochs affect training?

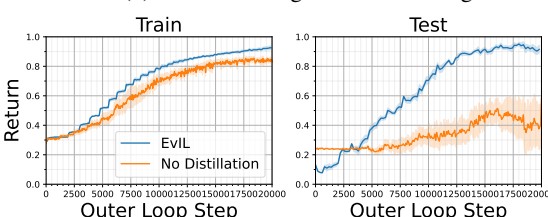

(a) Increasing inner loop steps  (b) Ablation of regularisation strategies

Figure 4: **Ablations** On the left, we show how gradually increasing the number of inner loop steps improves performance. When the inner loop steps are increased, we start from 1 inner loop step and repeat N times before increasing by one. In the plot, "None" corresponds to fully training the agent at each outer loop step. On the right, we show the impact of using distillation rather than simply adding $\ell_1$ regularisation to the ES objective.

**Distillation**: In Figure 4b we show that applying the analytical gradient of the L1 regularisation term through a separate distillation step is beneficial to both train and test performance, as ES struggles to co-optimise the BC and regularisation objectives.

**Increasing Inner Loop Steps**: In Figure 4a we show how gradually increasing the number of inner loop steps leads to better performance than fully training the inner loop at each outer loop step, which leads to vanishing meta-gradients.

**Warm-Starting Epochs**: To minimise interaction with the environment it is common to keep optimising the same inner loop policy $\pi$, rather than restarting it from a random initialisation for every inner loop. We found that always optimising the previous policy leads to very unstable training, and complete failure to recover the original reward function in most cases. However, we found a way to address this: We first start from the same policy, e.g. $\pi_0$, for hundreds of outer loop steps and only then use the final policy at the end of the inner loop, e.g. $\pi_1$, as a new initial policy for the subsequent outer loop steps. Overall, balances stability vs speed and converges faster than always re-initialising the agent policy from scratch.

## 6 CONCLUSION

**Summary.** We present EvIL, a general IL framework able to replicate an expert agent behaviour while simultaneously recover reward function and transition dynamics. Our framework can optimise any reward function, even non-differentiable ones, making our framework more flexible and widely applicable than previous methods. We show EvIL is better able to predict expert behaviour under changing environment dynamics that a traditional IRL method.

**Limitations.** Evolution-based methods can be sample inefficient. Our work makes heavy use of JAX-based simulators and algorithms to rapidly perform ES. Our method would likely struggle to scale to slow and complex simulators or other scenarios where environment interactions are expensive.

**Future Work.** Our inner loop optimisation procedure is repeatedly solving a slightly different RL optimisation problem at every outer loop step. This seems suboptimal, and other work in the area (Swamy et al., 2023) has demonstrated how using the state distribution of the expert can speed up the RL subroutine, by alleviating the exploration cost. ES could also help in this aspect by shaping the reward function to one that is easier and faster to learn, or learning a generator that can reset the training state to particularly useful states, limiting exploration. Additionally, our method could be applied to a multi-agent setting where, additionally to the current setting, agents' beliefs about other agents could be recovered, as well as specific training conditions that lead to a certain equilibrium among agents (Waugh et al., 2013). Finally, given the strong theoretical conditions required for imitation under causal confounding (Zhang et al., 2020; Swamy et al., 2022a;b), it would be interesting if evolution presented a more practically applicable solution.

**Reproducibility.**    As well as including hyperparameters in the Appendix, we commit to fully releasing our code on Github to ensure reproducibility. By using JAX, a specific run's results are reproducible deterministally given the same seed.

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
