# A   APPENDIX

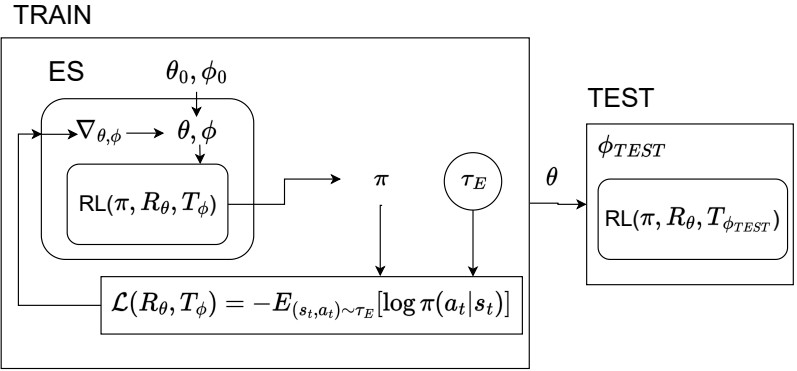

Figure 5: The EvIL framework

## A.1   ADDITIONAL OFFLINE RESULTS

In Figure 6 we show how we are able to increase performance in the Offline setting as more expert trajectories are provided. Interestingly, the Cross Entropy is lower for the case where only 2 trajectories are provided, although performance is much worse in terms of return.

## A.2   EXTENDED DISCUSSION ON LIMITATIONS

Although co-optimising the full transition can help generalise to unseen states in the train environment, it is unlikely to offer any benefits in a completely new test environment. Moreover, the reward function learnt on the optimised dynamics is unlikely to generalise to the test setting. However, if we have a strong prior over the space of possible transition functions (e.g. through a simulator) one can easily use EvIL to recover environment parameters and a transferable reward function that can be used at test time (e.g. on the same simulator but with different parameters).

In practice, we observe that transition functions recovered by EvIL are generally not interpretable, and don't produce transitions similar to the original environment. This is entirely to be expected given no prior knowledge of the environment.

## A.3   ENVIRONMENTS

The Reacher environment implementation can be found in the Gymnax Github (Lange, 2022). The functionality matches the classic MuJoCo (Todorov et al., 2012) environment, where a two-jointed robot arm moves with the goal of reaching a target. The target spawns at a random position at the beginning of each episode. The original reward for this environment is based on the distance between the robot's fingertip (the end of its effector) and the target. The Gridworld environment is a 5x5 grid with 2 different targets. The targets need to be found in a specific order for the reward to be returned (in all other cases it is 0). The episode ends when either both rewards have been found or the maximum number of timesteps (30) is reached.

## A.4   AIRL IMPLEMENTATION DETAILS

Our AIRL implementation follows the one described in the original AIRL paper, with the addition of gradient penalty (Gulrajani et al., 2017) to stabilise training. In all our AIRL runs, in each inner loop the policy is restarted from the last trained policy in the previous inner loop.

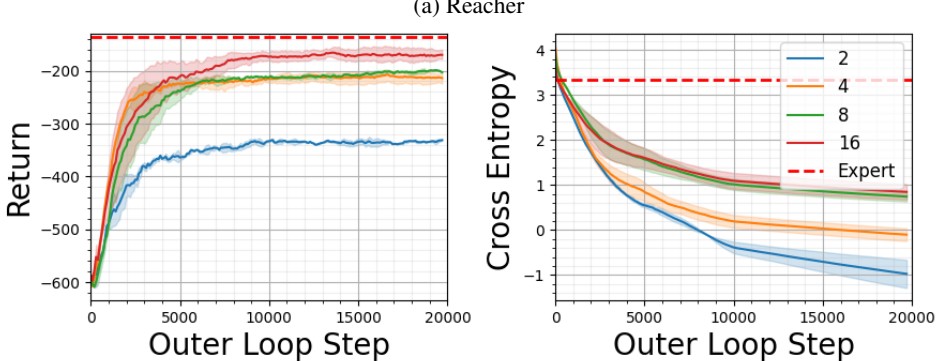

Figure 6: Comparison of Offline EvIL performance in the Reacher environment on a varying number of expert trajectories provided. Return is displayed on the left and Cross Entropy loss on the right.

Table 1: Hyperparameters for Training AIRL

| Parameter | Value |
|---|---|
| Number of Reward Hidden Layers | 2 |
| Size of Reward Hidden Layer | 256 |
| Reward Activation | tanh |
| Number of Outer Loop Steps | 20000 |
| Inner Loop Updates | 1 |
| Gradient Penalty Coefficient | 10 |
| Outer Loop Learning Rate Start | 1e-4 |
| Outer Loop Learning Rate Schedule | Linear |
| Outer Loop Learning Rate End | 1e-5 |

Table 2: Important parameters for Training EvIL on a 5x5 Gridworld

| Parameter | Value |
|---|---|
| Population Size | 128 |
| Number of Reward Hidden Layers | 2 |
| Size of Reward Hidden Layer | 256 |
| Reward Activation | tanh |
| Number of Generations | 20000 |
| ES Sigma Init | 0.03 |
| ES Sigma Decay | 1.00 |
| Inner Loop LR | 1e-3 |
| Number of Minibatches | 4 |
| Inner Loop Updates Start | 1 |
| Increase Inner Loop Updates Every | 1000 |
| Number of Update Epochs | 8 |
| Outer Loop Learning Rate | 7e-3 |
| Number of Environments | 256 |
| Reward Activation | tanh |

## A.5 HYPERPARAMETERS

In all of our experiments, we parameterise the reward function with a neural network with two hidden layers of size 256 and in the inner loop, policies are trained using PPO (Schulman et al., 2017). To minimise variance, we provide the same training seed to each population member within a generation, but change seeds across generations to ensure different environment initialisation are observed during training.

Table 3: Hyperparameters for BC Training

| Parameter | Value |
|---|---|
| Network Hidden Layers | 2 |
| Size of Network Hidden Layer | 256 |
| Network Activation | tanh |
| Number of Steps | 10000 |
| Num batches | 32 |
| Learning Rate | 1e-3 |

Table 4: Important parameters for Training EvIL Offline on Reacher

| Parameter | Value |
|---|---|
| Population Size | 128 |
| Number of Reward Hidden Layers | 2 |
| Size of Reward Hidden Layer | 256 |
| Number of Transition Dynamics Hidden Layers | 2 |
| Size of Transition Dynamics Hidden Layer | 256 |
| Number of Generations | 10000 |
| ES Sigma Init | 0.03 |
| ES Sigma Decay | 1.00 |
| Inner Loop LR | 0.004 |
| Inner Loop Updates Start | 1 |
| Increase Inner Loop Updates Every | 500 |
| Number of Minibatches | 10 |
| Number of Update Epochs | 10 |
| Outer Loop Learning Rate | 7e-3 |
| Number of Environments | 256 |
| Batch Size | 1024 |
| Reward Activation | ReLU |