# OpenReview forum: "EvIL: Evolution Strategies for Generalisable Imitation Learning"
_ICLR.cc/2024/Conference — Submitted to ICLR 2024_

### Official Review · Reviewer_foS1 · 2023-10-26

**Soundness:** 1 poor
**Presentation:** 2 fair
**Contribution:** 1 poor
**Rating:** 3
**Confidence:** 5

**Summary:**

This paper presents a method of model-based imitation learning based on learning and evolution. Specifically, the proposed method named EvIL is formulated as a bilevel optimization problem. The outer loop optimizes the reward and transition functions that are used to train a policy by reinforcement learning in the inner loop. The objective function of the outer loop is behavioral cloning loss, and CMA-ES is adopted to optimize the reward and transition functions. The evolved reward function is distilled to simplify the network architecture of the reward function, which improves the performance in the test environment. EvIL is evaluated on Gridworld and Reacher and outperforms BC and AIRL.

**Strengths:**

The proposed method is general and applicable to various policy optimization problems. The finding that using distillation is better than simply adding $\ell_1$ regularization to a loss function seems interesting. The experimental results are promising. In particular, transferability to unseen environments is interesting.

**Weaknesses:**

The literature review is insufficient. For example, the structure of EvIL is similar to the framework of learning and evolution. For example, Elfwing et al. (2018) optimize hyperparameters of RL by a simple evolutionary algorithm in the outer loop while the policy is trained by RL. In Nielum et al. (2010), the reward function is optimized by genetic programming. Please discuss the relation between EvIL and these classical studies. In addition, a more detailed survey on meta-learning is needed. For example, Evolved Policy Gradient (Houthooft et al., 2018) optimizes a differentiable loss function, where a similar evolutionary strategy is used in the outer loop.
S. Niekum et al. (2010). Genetic Programming for Reward Function Search. IEEE Transactions on Autonomous Mental Development. 2(2): 83-90.
S. Elfwing et al. (2018). Online Meta-Learning by Parallel Algorithm Competition. In Proc. of GECCO.
R. Houthooft et al. (2018). Evolved Policy Gradients. NeurIPS.

My second concern is the scalability of the algorithm. Since a naive evolutionary strategy is adopted to optimize the reward and transition functions, applying the proposed method to problems with a high-dimensional state space is problematic.

Finally, it would be better to re-organize Section 5 because Figure 3 is explained after Figure 2.

**Questions:**

1. Figure 1 seems interesting, but I do not fully understand the experimental conditions. The first paragraph of Section 4.1 mentions that the reward function optimized in the training environment is used to train policies in the test environment. However, Figure 1 shows that the reward and transition functions are optimized in the test environment. Please clarify the motivation for the experiments.
2. Figure 1 shows that the performance of EvIL is worse than that of AIRL in the training environment. However, AIRL failed to improve its performance in the test environment. Does it imply that the initial reward is tuned for the training environment?
3. Discussions about Figure 2 are needed. For example, in my view, there are no significant differences among EvIL, EvIL with supervised loss, and EvIL with supervised model. In addition, the performance in the online setting is much better than those of other methods. It implies that EvIL failed to find a good transition function even in the Poitmass environment. Would you discuss the quality of the optimized transition function in detail?
4. Figure 3 is interesting, but I am unsure how the features such as the obstacle position are retrieved from the reward function. As far as I understand, the reward function is represented by a neural network, which is a mapping from a state to a scalar value. Would you clarify the method to retrieve the reward features?

---

> ### Author Response · Authors · 2023-11-23
>
> We sincerely appreciate the reviewer's insightful feedback and their thorough understanding of the relevant literature. We are particularly encouraged by the recognition of our approach's generality and wide applicability.
>
> Addressing the reviewer's concerns:
>
> - While we acknowledge that our current experiments do not exhibit strong scalability, it is important to note that IRL is typically evaluated on MuJoCo environments, which have a smaller observation space compared to some of the environments we employed. Within these environments, we achieved performance parity with the AIRL baseline.
> - The papers mentioned have been added to the background under the Evolution Strategies subsection (2.1). While Niekum et al. (2010) do utilize evolution strategies to train a reward function, their objective is to enhance RL performance. In our approach, this is not feasible due to the absence of a reward signal. Similarly, Elfwing et al. (2018) also optimize for a reward function, but our framework operates under the assumption that no reward function is available. Evolved Policy Gradient (Houthooft et al., 2018) employs a similar setup with an ES outer loop optimizing for rewards  in the inner loop. However, the optimization targets differ between the outer and inner loops in both approaches. To summarise, our approach is fundamentally different for all the papers above as we don’t assume access to a reward function signal we can optimise, but instead we optimise for a cross entropy loss against the expert’s trajectories.
>
> Addressing the reviewer's specific questions:
>
> - The right-side plots in Figure 1 illustrate how the reward function generalizes to the test environment during optimization on the training data. Notably, the reward function is never optimized on the test data or environment; rather, we solely evaluate its performance there.
> - The original reward function that both AIRL and EvIL strive to recover remains the same. However, the reward function recovered by EvIL exhibits superior generalization to the test environment.
> - The plot demonstrates that EvIL, when combined with some knowledge of the reward function (as is often the case in real-world problems), can enhance performance in the offline setting. This is shown in Figure 2a.
> In Figure 2b, we show the performance of Offline Evil, here the transition function is learned only through Evolution Strategies and a Cross Entropy loss between the trained agent and the expert trajectories. This suggests our ability to recover transition dynamics from the data.
> In Figure 2d, we show the performance of EvIL where the dynamics model is learnt through supervised loss.
> Figure 2f, where we show the performance of Online EvIL, is not directly comparable to the other plots, as the optimisation process has access to more information (i.e. the environment).
> Offline EvIL did not match the performance of the expert (and Online EvIL) as it was only trained on a single expert trajectory. We have additional results on the Reacher environment (with more expert data) that intuitively demonstrate improved performance as the number of expert trajectories increases. The results have been added to the Appendix.
> - In the experiment depicted in Figure 4, we employ a partially known transition function and aim to uncover the remaining unknown variables. Our results demonstrate the successful recovery of these unknown variables, including obstacle position and size. Therefore it is the transition function, rather than the reward function, that serves as the source of this information.

---

### Official Review · Reviewer_4scx · 2023-10-31

**Soundness:** 3 good
**Presentation:** 3 good
**Contribution:** 3 good
**Rating:** 6
**Confidence:** 3

**Summary:**

This paper presents an evaluation strategy approach for immitation learning. This meta-learning approach optimizes the parameters of a reward function and environment dynamics in the outer loop, and then uses them in the inner loop to reinforcement learn. The authors show that their approach is able to learn the reward function induced by an expert bot behaviour, also when the dynamics of the environment are not fully known. The method is evaluated on gridworld, reacher, and pointmass environment and compared to a standard behavioural cloning approach and adverserial inverse RL.

**Strengths:**

- The paper presents an interesting application of ES, taking advantage of the fact that neither the reward nor the training - procedure need
to differentiable.
- The approach addresses some key limitations of most IL approaches
- Baseline comparisons and ablation tests are sensible
- Promising results in three benchmarks

**Weaknesses:**

- Some acronomys are not defined, e.g. AIRL (adverserial inverse reinforcement learning). What are its training setup?
- Approach should be evaluated on more environments and potentially complexer ones. Is the approach generally better to BC or does it
depend on the type of environment?
- It would be useful to test how transferable the learned reward functions are to similar domains
- Hyperparaemters for some of the experiment steps don’t seem to be mentioned (e.g. learning rates, etc.?)
- The environments are not described in the paper. Details should at least be added to the appendix.

Minor comments:
Typo page 3: "the the expert demonstration"

**Questions:**

- Why were these particular evaluation test environments chosen?
- What are the parameters for the BC training?

---

> ### Author Response · Authors · 2023-11-23
>
> We wholeheartedly appreciate the reviewer's encouraging feedback and their recognition of the intriguing application of our approach. We are delighted that the reviewer acknowledged the advantages of utilizing non-differentiable rewards and training methods.
>
> Addressing the reviewer's concerns:
> - We have addressed the reviewer's suggestions by including the BC hyperparameters in the appendix, defining the acronyms, and providing detailed descriptions of the environments.
> - The transferability and generalizability of the recovered reward function is what we are testing for in the test version of our environment (e.g. different joint dampening in Reacher and a maze vs no-maze setting in the Gridworld)
> - AIRL stands for Adversarial Inverse Reinforcement Learning, which is essentially equivalent to Generative Adversarial Inverse Reinforcement Learning (GAIL). In AIRL/GAIL, there are two players: a discriminator (used as reward function) and a player. The reward function player tries to pick out differences between the current learner policy and the expert demonstration, while the policy player attempts to maximise this reward function to move closer to expert behaviour. This setup is effectively a GAN. For a more theoretical explanation, please refer to [1].
> - We are currently in the process of expanding our method to MuJoCo tasks by changing environment parameters like link lengths and joint stiffness to generate a diverse range of dynamics. This approach generally surpasses BC due to its foundation in Inverse Reinforcement Learning (training the agent in the inner loop) instead of simply imitating the expert's actions. BC has been shown to suffer from compounding errors and poor generalization.
>
> To address the reviewer's questions:
>
> - The Gridworld and Reacher environments were chosen because they pose a non-trivial challenge in recovering the reward function. The objective in these environments is to change position at the beginning of each episode, unlike other environments like CartPole which rely solely on a positive reward signal.
>
> [1] Swamy, Gokul, et al. "Of moments and matching: A game-theoretic framework for closing the imitation gap." International Conference on Machine Learning. PMLR, 2021.

---

### Official Review · Reviewer_vCw2 · 2023-11-04

**Soundness:** 2 fair
**Presentation:** 1 poor
**Contribution:** 1 poor
**Rating:** 3
**Confidence:** 4

**Summary:**

This paper presents a way of applying evolutionary optimization to inverse reinforcement learning. The performance of the method is evaluated in some toy problems and compared with AIRL.

**Strengths:**

It is certain that evolution can provide a solutions to any problem.
Originality of the proposal includes:
• estimating the gradients of policy and transition model parameters with Gaussian mutation and perform gradient decent, rather than usual selection.
• tuning of inner-loop steps to avoid disappearance of the above gradient
• L1 distillation of the estimated reward functions

**Weaknesses:**

A crucial issue in evolutionary approach is how practically and competitively a real-world problem can be solved.
Figure 1 presents the comparison with AIRL, but the implementation of AIRL is not sufficiently documented.
The x axis is the number of outer loops, but how many interactions with the environment happened in each outer loop for EvIL and AIRL?
The basic parameters like the population size N should be reported in the main text.

**Questions:**

How does the gradient approach compare with standard selection approach in evolutionary optimization? This can be benchmarked in addition to the ablation study within the EvIL framework.
4.1, 2.: The link between the non-differentiable objectives and potential-based shaping is not clear to me.
Page 2, para. 3: environment dynamics are trained: did you mean dynamics models are trained? You can train your agent but not the environment.
What were the target reward functions of each benchmark? It is not clear even in the supplementary materials. For each goal point, how sparsely was the reward functions set?

---

> ### Author Response · Authors · 2023-11-23
>
> We express our gratitude to the reviewer for their valuable feedback and for recognizing the novel contributions of our work, particularly the L1 distillation technique and the dynamic adjustment of the number of inner loop gradient updates to prevent gradient vanishing issues.
>
> Addressing the reviewer's concerns:
>
> - While the number of inner loop steps for both AIRL and EvIL is the same, in the EvIL case, we leverage the power of parallel computation by training 128 agents simultaneously. This parallel training strategy significantly enhances the computational efficiency of our approach.
> - To address the reviewer's request, a description of the AIRL implementation, along with the corresponding hyperparameters, has been included in the Appendix.
>
> Question responses:
>
> - The comparison with gradient approaches is the one we did by comparing with AIRL, the standard approach in this case
>
> - Optimizing a potential-based shaping function using conventional meta-learning methods poses a significant challenge. As potential-based shaping only affects training speed, and not the optimal policy, it cannot be optimised with first-order methods. EvIL presents a perfect solution, as it is a zeroth-order optimisation method. The shaping objective could be optimized concurrently with the standard Cross-Entropy objective, resulting in better-shaped reward functions that are faster to optimize.
> - Yes, we meant the dynamics models are trained, this has been fixed in the original paper
> - The reward function for the Reacher environment is the standard one available in Gymnax and Brax, which utilizes a continuous reward signal based on the distance between the agent's end effector (“fingertip”) and the goal position. In the Gridworld environment, a reward of 1 is granted if all goals are discovered in the correct sequence before the episode ends; otherwise, the reward is 0. Similarly, in the goal point environment, the reward is only provided when the agent reaches the goal. This description has been added to the paper’s Appendix.

---

### Official Review · Reviewer_p69Z · 2023-11-08

**Soundness:** 2 fair
**Presentation:** 2 fair
**Contribution:** 2 fair
**Rating:** 3
**Confidence:** 3

**Summary:**

This paper introduces Evolutionary Imitation Learning (EvIL), a versatile strategy for imitation learning (IL) that can forecast the behavior of agents in environments across different dynamics.
EvIL employs Evolution Strategies to concurrently meta-optimize the parameters (such as reward functions and dynamics) that are input into a subsequent reinforcement learning process. The authors claim that they are able to inherit some of the benefits of IRL and
In addition, since they are using a population-based method, the reward or the training process does not need to be differentiable. They validate the robustness and generalization ability through some simple environments.  The results show that they can perform better than BC and classic IRL approaches.

**Strengths:**

- The ability to predict an expert’s behavior in an environment different from the one where the trajectory data was collected is an important topic
- Able to handle undifferentiable rewards.

**Weaknesses:**

- Since the paper only uses experiments to demonstrate the method's effectiveness, I believe that the experiments are too few and too simple. Usually, when someone talks about the generalization of RL regarding the transition function, I will think of more challenging and practical modifications like different gravity, mass, and friction.
- The method is very slow since it involves multiple rounds of training.
- The authors do not clarify why we should use their design since the method is not the most straightforward way to solve the problem of previous work.  An easy way to focus on only one scenario first and talk about why we need the method in that scenario (e.g., online).

**Questions:**

- It seems like in expert demonstrations you only use one kind of transition function is that correct?

---

> ### Author Response · Authors · 2023-11-23
>
> We sincerely appreciate the reviewer's comprehensive review of our work.
>
> Addressing the reviewer's concerns:
>
> - We are currently engaged in expanding our method's applicability to MuJoCo tasks, where we intend to introduce variations in environment parameters such as link lengths and joint stiffness to create a diverse range of dynamics.
> - The iterative nature of our method, involving multiple rounds of training, aligns with standard practice for IRL methods. Both primal IRL approaches like GAIL/AIRL and dual IRL techniques like MaxEnt IRL are iterative procedures. Additionally, our method is not slower than standard IRL approaches, but is comparable due to the massive parallelization we are able to achieve via JAX. As an example, the PureJaxRL library archives a 1000x speed up over other standard PyTorch RL implementations.
> - As you rightly pointed out, our method's strength lies in its ability to optimize non-differentiable reward functions through a non-differentiable optimization process. Moreover, we emphasize behavioral prediction outside the training distribution and demonstrate our method's superior performance compared to standard AIRL. While AIRL performs satisfactorily within the training environment, our experiments highlight our ability to achieve parity with traditional methods while expanding the range of applicable reward model families. For instance, we can optimize for symbolic expressions of environment features, as demonstrated in the recent EUREKA paper [1]. Additionally, we show better performance in the test environments, indicating rewards recovered by EvIL generalize better.
>
> Question response:
>
> - Yes, the expert trajectories are indeed sampled from a single environment.
>
> [1] Ma, Yecheng Jason, et al. "Eureka: Human-Level Reward Design via Coding Large Language Models." arXiv preprint arXiv:2310.12931 (2023)
>
> [2] Lu, Chris Xiaoxuan et al. “Discovered Policy Optimisation.” ArXiv abs/2210.05639 (2022)

---

### Meta-Review · Area_Chair_fMzt · 2023-12-14

**Metareview:**

The paper introduces Evolutionary Imitation Learning (EvIL), for both tabular and continuous control settings. By optimizing the current reward function through reinforcement learning, EvIL uses function approximators such as deep networks to represent the reward function and policy. The authors commit to reproducibility by including hyperparameters in the Appendix and releasing their code on Github. ES methods typically don't go with theoretical analysis. The experimental results are very limited, without sufficient comparisons with representative IL methods. The speed and scalability should also be discussed.

**Justification For Why Not Higher Score:**

No theoretical analysis (kind of acceptable for ES methods). The experimental results are very limited, without sufficient comparisons with representative IL methods. Need more justification on the advantage of proposed method.

**Justification For Why Not Lower Score:**

N/A

---

### Decision · Program_Chairs · 2024-01-16

Reject